# Advanced Video-Based Processing for Low-Cost Damage Assessment of Buildings under Seismic Loading in Shaking Table Tests

**DOI:** 10.3390/s23115303

**Published:** 2023-06-02

**Authors:** Antonino Cataldo, Ivan Roselli, Vincenzo Fioriti, Fernando Saitta, Alessandro Colucci, Angelo Tatì, Felice Carlo Ponzo, Rocco Ditommaso, Canio Mennuti, Alessandro Marzani

**Affiliations:** 1ENEA—Italian National Agency for New Technologies, Energy and Sustainable Economic Development, 00196 Rome, Italy; 2Scuola di Ingegneria, University of Basilicata, 85100 Potenza, Italy; 3INAIL—Istituto Nazionale Assicurazione Contro gli Infortuni sul Lavoro, 00144 Rome, Italy; 4Dipartimento di Ingegneria Civile, Chimica, Ambientale e dei Materiali, University of Bologna, 85100 Potenza, Italy

**Keywords:** video based, vibrational monitoring, structural damage, low-cost monitoring methods, modal parameters identification, damage location

## Abstract

This paper explores the potential of a low-cost, advanced video-based technique for the assessment of structural damage to buildings caused by seismic loading. A low-cost, high-speed video camera was utilized for the motion magnification processing of footage of a two-story reinforced-concrete frame building subjected to shaking table tests. The damage after seismic loading was estimated by analyzing the dynamic behavior (i.e., modal parameters) and the structural deformations of the building in magnified videos. The results using the motion magnification procedure were compared for validation of the method of the damage assessment obtained through analyses of conventional accelerometric sensors and high-precision optical markers tracked using a passive 3D motion capture system. In addition, 3D laser scanning to obtain an accurate survey of the building geometry before and after the seismic tests was carried out. In particular, accelerometric recordings were also processed and analyzed using several stationary and nonstationary signal processing techniques with the aim of analyzing the linear behavior of the undamaged structure and the nonlinear structural behavior during damaging shaking table tests. The proposed procedure based on the analysis of magnified videos provided an accurate estimate of the main modal frequency and the damage location through the analysis of the modal shapes, which were confirmed using advanced analyses of the accelerometric data. Consequently, the main novelty of the study was the highlighting of a simple procedure with high potential for the extraction and analysis of modal parameters, with a special focus on the analysis of the modal shape’s curvature, which provides accurate information on the location of the damage in a structure, while using a noncontact and low-cost method.

## 1. Introduction

Several techniques for seismic monitoring are available, and multiple methods and sensors are used. They are generally intended to sense the vibrations that occur in a structure or the possible changes in the physical parameters related to the structural health as a consequence of a seismic event. In fact, several types of earthquakes may occur depending on the site where the structure is located. In particular, sudden and unexpected high amplitude earthquakes, or so-called high activity earthquake swarm events, can occur, potentially causing damage to civil engineering structures [1].

In recent years, the application of image-based techniques to structural monitoring has gained increasing interest [2,3,4]. One the one hand, digital image and video processing methods and vision technologies have seen remarkable advancements at more affordable costs thanks to the development of advanced materials for hardware equipment and improvements in new algorithms for more powerful and robust processing [5,6,7,8,9]. On the other hand, conventional methods for structural monitoring usually require a rather difficult and costly equipment set-up, requiring the positioning of a relatively limited number of expensive sensors (e.g., accelerometers and velocimeters) as the measurement points of the structure, which need to be physically reachable by human operators, sometimes even leading to safety concerns [10]. In addition, these conventional contact methods usually require periodical maintenance and potential replacement, which causes additional costs with the passage of time. Moreover, using many contact sensors adds, by definition, extra mass loading to the monitored structure, which can have an obvious negative impact on lightweight structures. Furthermore, when using a limited number of sensors, it becomes crucial to locate them at proper positions on the structure to optimize the response. To this aim, a great deal of research has been conducted on optimal sensor placement using a variety of placement techniques and criteria [11]. Alternative, noncontact methods can be used with the advantage of eliminating the challenges of contact-based sensors. Recent advances in technology provide several tools for contactless sensing, which is overtaking traditional contact sensing. Among the emergent approaches on noncontact methods and tools, the most relevant techniques are vision-based, wireless, embedded sensors, and guided waves [12]. In particular, video-based methods such as motion capture systems (MCS) that require the use of fiducial markers proved very effective in laboratory tests [13,14] but have substantially similar issues for marker positioning, even if the markers are typically cheaper than conventional sensors. This issue is overcome completely using video-based methods that are able to monitor the whole structure or most of it without fiducial markers. For the above reasons, improving the capability to extract parameters related to structural health from footage taken at a distance by video cameras may play a crucial role in the low-cost and safe monitoring of structures. Video-based methods include applications where unmanned aerial vehicles (UAVs) are employed [15,16]. UAV methods for damage detection and quantification mainly focus on the high-resolution visual inspection of structures that allow for the accurate survey of cracks and defects. In addition, UAV-based approaches were explored for structural identification, but they need the development of effective algorithms to compensate for the effect of the UAV’s egomotion in the displacement signals, which is certainly a remarkable complication in respect to the use of fixed cameras. Some video-based methods deal with the problem of the need for numerous measurement points. Among such methods, interesting possibilities are provided by optical systems that can perform such measurements of even very big objects (e.g., buildings), including video recorders, virtual vision sensors, digital image correlation (DIC), and 3D laser Doppler vibrometry [17,18]. In particular, an innovative system utilizing a 3D laser Doppler vibration measurement system was recently proposed [18]. This system is able to extract the modal parameters of civil structures or buildings’ structural elements with good accuracy and a reduced required time. However, the 3D laser Doppler vibrometry still requires the use of quite expensive equipment with respect to low-cost cameras.

So far, conventional methods based on accelerometers and seismographs still provide, undoubtedly, results with much higher accuracy and reliability than video-based methods. Moreover, it is worth noting that for real structures the presence of nonstructural components, such as external plaster and external coat and finishes, could strongly modify results retrieved using only noncontact sensor-based methods, especially to characterize the structural motion in critical conditions.

However, video-based methods can contribute to managing structures and infrastructures with more flexible and affordable monitoring systems.

In this context, particularly interesting are the potentialities of the motion magnification (MM) method [19] and the related advanced processing techniques [20], which permit the amplification and analysis of the imperceptible movements of objects in a video. In fact, most conventional methods for structural health monitoring (SHM) are based on sensors that are able to record movements in a structure, e.g., static deformations or ambient vibrations [21]. In particular, the analysis of ambient vibration motions is widely used to perform the modal analysis of a structure, as the modal parameters can be easily related to the evolution of the state of damage of buildings [22]. This has essentially motivated the recent remarkable efforts to improve the MM algorithms and the related video processing methods to extract the modal parameters from footage of structures. Some authors explored the potentialities of MM video-based methods for modal analysis using high-speed, high-resolution, and high-cost cameras, obtaining encouraging results in laboratory experiments [23]. Subsequently, the use of low-cost equipment and consumer-grade cameras (CGCs) was also explored both in laboratory and outdoor environments, with less accurate but still interesting results in terms of a rough estimate of the main modal frequency [24]. Such studies demonstrate that low-speed cameras (typically at 24–30 fps) can actually detect modal frequencies lower than 3 Hz, even in an outdoor environment. Such low frequencies are common for the main modal frequency of most structural typologies. However, the quality of the used equipment still plays a decisive role, in particular in terms of the video resolution and speed but also the camera’s dynamic range and optics are quite important in relation to the lighting conditions during the acquisitions [24]. It is also worth noting that by now some common smartphones and CGCs already offer the possibility to take videos at 1080p or even 4K resolution standards at 60 fps or even 120 fps with a cost of USD 1 k or less.

Nonetheless, the present study aimed to go a step further by extracting also the modal shape associated with the identified modal frequency via MM analysis of videos taken with relatively low-cost and low-quality equipment. The interest in monitoring the modal shape is that its changes in curvature can be used to locate the damage in a structure with higher accuracy than operational deformed shapes [25,26]. In particular, the modal curvature has been successfully used to locate damage arising in framed structures [27]. In such studies, though, the curvature was defined through inter-story deformations with the aim of identifying the most damaged floors, while video-based methods can generalize these concepts to locate damage at any point in the structure.

The proposed MM–based procedure was validated using the experimental application of a two-thirds scale, two-story mockup of a reinforced-concrete (RC) framed building representative of a widely common construction typology of the Italian territory. The mockup was tested on a shaking table, reproducing a recent Italian earthquake.

In order to validate the results obtained using the proposed procedure, a variety of other sensors and methods were also used in the experiment. In particular, the building was instrumented with conventional accelerometers and high-precision optical markers tracked by a passive 3D MCS. In addition, the building geometry was accurately surveyed with 3D laser scanning before and after the seismic tests.

Accelerometric recordings were processed and analyzed using several stationary and nonstationary techniques with the aim of evaluating the linear behavior of the undamaged structure and the nonlinear structural behavior in response to damage during shaking table tests. In terms of the linear and nonlinear modal parameters, the results obtained using the band variable filter [27], based on the Stockwell transform [28,29] and the short–time impulse response function (STIRF) [30], highlighted the importance of using time-frequency analyses to study the response to structural damage, as well as with the aim of better evaluating the health condition of the monitored structure.

## 2. Proposed MM-Based Procedure

The proposed procedure comprises different processing steps. In Figure 1, the essential flowchart of the proposed procedure is illustrated.

A video camera records footage of only a few minutes of the object. In the second step, the footage is processed using a software tool provided by the Computer Science and Artificial Intelligence Laboratory at the Massachusetts Institute of Technology (MIT–CSAIL). The phase-based MM version (PBMM) of the algorithm is implemented with this software is used [31]. The main parameters to be set are the frequency range and the amplification factor.

The frequency range indicates the group of frequencies whose magnification will be enhanced with respect to the others. However, the precise specifications of their tuning are lacking, as warned by the MIT researchers themselves.

Therefore, it is advisable to collect as much information as possible on the studied object, especially in terms of the frequency range of interest, using a frequency range and required level of amplification within that range. An excessive amplification or too large frequency range produces disturbing image distortion and noise. Therefore, a trade-off between noise and amplification is always necessary. The duration (T) of the video is inversely correlated to the frequency resolution in the frequency domain: the larger the T, the better the frequency resolution. Unfortunately, a large T means a much longer elaboration time. Moreover, taking into account the Shannon–Nyquist theorem, the camera frame rate (fps) becomes an important parameter. It is suggested, however, to adopt a significant data oversampling to compensate for possible aliasing phenomena.

Then, an appropriate region-of-interest (ROI) is selected for analysis in the frequency domain. In fact, the whole full-resolution image could be theoretically processed, but it would be computationally uselessly heavy, as not all pixels in the image provide equally significant information. So, the choice of an efficient ROI is crucial for an efficient identification of the modal frequencies. On the one hand, the ROI should contain a portion of the studied structure with the highest image contrast with respect to the background so that structural motion is captured with the highest accuracy. To this purpose, an image entropy criterion can provide a useful indication [32]. On the other hand, the best results can be obtained by points of the structure, with the highest expected displacements associated to the considered modal shapes. Without going into details about the several existing optimal sensor placement approaches [12], the ROI selection can be treated following somewhat similar approaches. However, contrarily to conventional sensor placement, the ROI size, shape, and position can be easily changed at any time with great flexibility so that an optimal ROI can be found manually in a few attempts. Nonetheless, it is highly recommended to preliminarily calculate the expected mode shapes of the structure using finite element analysis (FEA), which provides useful indications to guide the ROI selection. Moreover, the ROI size should consider the available computational resources, as the larger the ROI, the higher the computational time. Finally, the ROI shape is a rectangle that should be as flat as possible in the direction orthogonal to the main direction of motion.

The values of the pixels extracted from the ROI are processed in the frequency domain, typically using power spectral density (PSD) to identify the main frequencies. Once the pixel time-series are extracted from the magnified video, they are averaged and processed just as the usual standard signals. The somewhat smooth appearance of the PSD depends on the short time span of the video that decreases the PSD frequency resolution. Instances of this procedure can be easily found in the literature, see for example [33].

After the identification of f_r_, a new MM video is produced with the passband filter centered on f_r_ and with a sufficiently large amplification factor (α) so that the MM video actually visualizes the structure moving according to the modal shape.

This latter MM video is then processed by searching for the frame with maximum displacement (FMD) and frame with zero displacement (FZD) of the studied structure. The FMD is assumed as the best frame to identify the modal shape, while the FZD represents the undeformed structure and is used as a reference. The FZD might be identified as a frame with minimum displacements, as no frame will possibly have zero displacements, given the unavoidable noise in the image.

At this point, the FMD and FZD undergo further processing. Preparatory image processing steps, which essentially make the image’s structural lines more evident, might include color adjustment and contrast maximization. Then, an inverted edge detection algorithm [34] is applied in order to enhance the main structural lines in the image (e.g., lines of columns and floor slabs). Such lines are extracted from both the FMD and FZD. The relative displacements between corresponding structural lines in the FMD and FZD are calculated to obtain a magnified modal shape (MMS), which can be demagnified by dividing by α to obtain an estimate of the real modal shape.

In order to verify the consistency of the results, the proposed method can be analogously applied to more couples of the frames corresponding to the relative maximum and minimum displacements in any period of the oscillation of the studied structure in the magnified video at a given modal frequency. For example, in a magnified video with a duration of 20 s at a modal frequency of 3 Hz (e.g., three oscillations per second), we have 60 oscillations; therefore, we can identify 60 couples of relative FMD and FZD frames. Finally, an averaged modal shape can be calculated and analyzed. However, this would imply a much greater computational time. Consequently, we limited the extraction of the modal shape to the three amplest oscillations just to demonstrate that the result obtained with only the absolute maximum FMD and minimum FZD frames is not an outliner.

## 3. Lab Experimentation

The experiment was carried out at the seismic hall of the ENEA Casaccia Research Center, near Rome, Italy, using a 4 m × 4 m, 6 degrees-of-freedom (DOFs) shaking table. The shaking table was used for seismic testing of a two-thirds scale, two-story mockup of a framed weakly RC building. It was designed in accordance with the Italian codes and standards of the 1960s and 1970s, when a large part of the Italian building stock was designed during the real estate “boom” of the post-war reconstruction, so that it nowadays represents one of the most widely common construction typologies on the Italian territory [35].

The building mockup was designed to resist only to vertical loads according to the Italian codes of that time.

As for the material properties, the utilized concrete had a characteristic compressive strength of 20 MPa, and the steel bars had a tensile characteristic yield strength of 450 MPa. Extracts of the design drawings and the dimensions of the tested mockup are illustrated in Figure 2. In addition, the slabs were typical of common Italian building stock, and they were built as one-way ribbed floor slabs (drawing of the design shown in Figure 3) and lightened in weight through the use of hollow clay bricks, and the floor slab weight was approximately 21 kN. The overall mass of the building was approximately 150 kN. An additional load of 12 kN was added to each floor by steel plates placed at nine positions in a 3 × 3 matrix. The configuration of the additional load is shown in Figure 4a, with slight eccentricity with respect to the building center of mass.

After positioning on the shaking table, the building’s base was fixed to the table and a 3D geometrical survey using a laser scanner was carried out. Four stations placed near each building column were acquired (Figure 4b). This acquisition was repeated after the end of the shaking table tests.

Several instruments were utilized to the acquire motion data of the structure positioned on the shaking table (Figure 4c). In particular, the positions of the MM camera, as well as the accelerometers and markers of an optical 3D MCS, are illustrated in Figure 5. For a nomenclature of the measurement points, the building columns were named from 1 to 4 counterclockwise, and the building facades were named S, E, N, and W according to cardinal points.

The camera used for the MM acquisitions had mounted a CMOS sensor Onsemi PYTHON 1300 with a full resolution of 1280 × 1024 pixels (1264 × 1016 pixels in the color mode) and a maximum frame rate of 210 fps with reduced resolution (Figure 6a).

The optical 3D MCS is based on Vicon technology with a constellation of 13 near-infrared (NIR) cameras positioned at safe distances around the shaking table (see Figure 6b). The 3D motion capture NIR cameras with 5 Mpixel at 200 fps, acquired the position of 68 markers located on the building mockup. The used markers were simple 25 mm diameter plastic balls covered with a retro-reflecting coating. They were placed at the base of each column, at the column beam joints, and in the center of both floors’ beams. The system measured the markers’ position with accuracies of approximately 0.03 mm and 0.15 mm in terms of the root mean square (RMS) error in static and dynamic (seismic) conditions, respectively. The used accelerometers were PCB 3701G3FA3G, whose broadband capacitive sensors had a 3 g peak measurement range. They were acquired at a sampling frequency of 200 Hz. Eleven accelerometer channels were positioned on each floor and at the specimen’s base, as shown in Figure 5a.

The seismic input was based on an earthquake that occurred on 30 October 2016, the largest event in the context of the 2016–2017 Central Italy seismic sequence, with an estimated magnitude of M_w_ 6.5 that struck 6 km north of Norcia at 07:40 local time (06:40 UTC). In particular, the shake recorded at the Savelli seismic station (NRC, Lat. 42.72492 Lon. 13.12578 WGS84, soil type A), located near Norcia (Figure 7), was considered. This was scaled in acceleration in order to obtain the NRC tests (see the shaking table sequence in Table 1), with an increasing step of 0.1 g of the peak ground acceleration (PGA) up to the natural shake of 0.87 g of the PGA. Each seismic test was intercalated with dynamic identification tests with the white noise frequency spectrum (also known as “random” tests) at 0.05 g of PGA (Rnd tests in Table 1). The seismic NRC_100 test was repeated to assess degradation cause by damage with same earthquake. The two random Rnd_10 and Rnd_11 tests were executed for checking the repeatability of the modal identification with MM using different camera parameters. All tests were triaxial.

## 4. Numerical Modeling and Analysis

The dynamic behavior of the structure was evaluated using finite element models to coherently select the seismic input for the shaking table test. The software Midas Gen was used for this purpose. Beam elements with six degrees of freedom per node were used. The floors were not directly modeled, assuming a rigid diaphragm behavior with mass, and their rotary inertia was evaluated separately and applied to the center of mass of each floor. For the vertical action, the mass of the floor was applied to the nodal point of the structure. The mass and inertia of the floor also included the additional point masses placed on the floors to simulate live loads: six of 100 kg and three of 200 kg.

The first analyses were conducted assuming the design values for the concrete Young’s modulus, taken from the standard *E_c_* = 29.96 GPa; subsequently, the model was updated considering *E_c_* = 27.95 GPa, derived from the experimental evaluation of the concrete strength.

However, the analysis of the experimental dynamic data from the first random test (Rnd_01), which was representative of the undamaged structure, showed much lower frequency values than those derived from the initial linear FE model (see Table 2). Instead, a good agreement was obtained by assuming a cracked flexural stiffness in the model, as is reasonable according to the technical codes for seismic analyses, even in the case of service limit states. Therefore, *E_c_* was finally assumed to be equal to 20 GPa, and a reduced inertia of the structural members was adopted. In more detail, we considered stiffness ratios of 0.7 for the columns and 0.64 and 0.4 for the *x*-directed and *y*-directed beams, respectively. The difference attributed to the beams is related to the different contributions of the one-way floor. Figure 8 shows the first three mode shapes and their associated frequencies after updating the linear model.

For the nonlinear analyses, a Mander constitutive model [36] was adopted for concrete and Menegotto-Pinto [37] for steel. Therefore, three zones characterized the concrete members: (i) not confined for the cover, (ii) confined for the core of the columns, and (iii) confined for the core of the beams. Uncracked initial stiffness was considered in this case. A concrete unconfined strength of *f_c_* = 22,217 MPa and a steel yielding strength of *f_y_* = 450 MPa were considered.

A seismic analysis was performed with the direct integration of the equations of motion using the earthquake recorded at the Savelli station in Norcia, 30 October 2016 (NRC_100). The final state of the plastic zones due to the rotation around the *y*-axis in the local reference of the element is shown in Figure 9. No collapse was to be expected, but the structure was expected to exhibit serious damage, according to the objectives of the test. Concerning the top of one of the columns, some of the details of the results are reported in Figure 10, showing the partial cracking of the concrete at the end of the analysis with a scaled input factor of 0.1, which approximately corresponds to NRC_01g.

## 5. Experimental Data Processing

### 5.1. MM–Based Video Processing

Only the videos acquired during the Rnd tests were taken into account, because the MM algorithm works well when the motion is very minimal. Videos in which the motion is strong, such as in seismic tests, are not appropriate for MM processing, as they induce abnormal distortion in the final MM video. However, here, the aim was to extract indications of damage from real structures subjected to ambient vibration, so Rnd tests are more representative of such conditions. In particular, Rnd_02, Rnd_06, and Rnd_11 were acquired in the HD mode (1264 × 1016) at 50 fps, while all other Rnd tests were acquired in the LD mode (632 × 508) at 150 fps.

After the preliminary FEA of the tested structure, four tentative ROIs, named with subscripts from A to D, were selected for the upper floor, where the simulated first bending mode in the x-direction provided the highest values of deformation (Figure 11a). The selected ROIs were assessed in terms of an image contrast by calculating the Michelson index *m*, which is defined as follows [38]:(1)$$m=\frac{I_{max}-I_{min}}{I_{max}+I_{min}}$$
where I_max_ and I_min_ are the highest and lowest pixel values in the ROI, respectively. Then, the ROIs’ pixels were analyzed in the frequency domain, and the PSD was calculated. As illustrated in Figure 11b, ROI_A_ has with the highest *m* value and provided a good identification of the modal frequency. In fact, the contrast in ROI_A_ was determined by the difference in color between the column edge and the white background, while the contrast in the other ROIs was very poor, as it was only because of the slight heterogeneities in the color of the concrete and the presence of markers. Thus, the final ROI was chosen, enlarging ROI_A_, as shown in Figure 12a. The PSD was then calculated for each Rnd test (Figure 12b).

An MM video was produced for each Rnd test with an FIR passband filter with the window centered on the identified first frequency and a bandwidth of ±0.2 Hz, while α was equal to 100. All MM–processed videos are available at the link in the Appendix A section. The MM videos were then processed by searching for the frame with the maximum mean value of the displacements within the ROI, which was assumed as the FMD of the MM video. The frame with the minimum mean value of displacements within the ROI was assumed as the FZD of the MM video. The FMDs and FZDs were processed using color adjustment and contrast maximization. Then, the invert edge detection algorithm was applied to outline the main structural lines of the columns and floor slabs (Figure 13a). In particular, the column lines were considered for the first modal shape identification, as it was expected to be an x-direction bending mode (Figure 13b). The column lines were digitized with 100 points (one point every 4 cm, for a total height of 400 cm). The relative displacements in the x-direction between the corresponding structural lines in the FMD and FZD were calculated to extract the modal shape (Figure 13c).

The extraction of the modal shape was repeated using the three amplest oscillations in the MM video to demonstrate that the result obtained, using only the absolute maximum FMD and minimum FZD frames, was not due to a possible outlier. The corresponding results are indicated with “1”, “2”, and “3”.

### 5.2. Optical 3D MCS Markers

The displacement data from the markers were processed for the extraction of the modal parameters and for the monitoring of the structural deformations. In particular, the markers data of the Rnd tests were processed using experimental modal analysis (EMA) techniques through the multi-input-multi-output (MIMO) method. In fact, the four markers at the column base were considered input signals, while all other markers on the structure were considered output signals. More specifically, the H_1_ estimator of the frequency response function (FRF) was calculated according to the following common formulation:(2)$$H_{1}\left(\omega \right)=\frac{S_{io}\left(\omega \right)}{S_{ii}\left(\omega \right)}$$
where S_io_ is the cross power spectrum of the input and output signals, respectively; S_ii_ is the auto power spectrum of the input signals; and ω is the angular frequency, calculated as 2π·f, where f is the frequency. The H_1_ estimator, obtained from the marker data, is then compared with the theoretical single-degree-of-freedom (SDOF) response by curve-fitting method for the accurate extraction of the resonance frequency (Figure 14).

In addition, the modal shape was calculated by filtering the markers’ displacements at the identified modal frequency. A 4th-order Butterworth passband filter with the window centered on the identified first frequency and a bandwidth of ±0.2 Hz was applied.

Relative displacements between markers in the x– and y-directions during the seismic tests were used to obtain the inter-story drift at the first and second floors with respect to the markers at the column bases. The inter-story drift can be defined as the relative translational displacement between two consecutive floors and is widely considered an important engineering demand parameter and indicator of structural performance [39].

The distances between markers were also investigated to detect structural deformations and, notably, a crack’s opening and width [40]. In particular, in Figure 15 the typical behavior of a crack’s opening during a seismic shake is depicted. Cracks typically open and close during a shake (i.e., dynamic condition) and eventually remain residually open at the end of the test (i.e., static condition) when permanent plastic deformation of the structure is achieved. Markers can detect a crack’s width in both dynamic and static conditions so as to accurately characterize the crack’s behavior.

### 5.3. Advanced Accelerometers Processing

In addition to the conventional EMA processing of the Rnd tests, the accelerometer data of the seismic tests were also processed using more advanced techniques. In particular, the nonlinear structural behavior of the mockup structure was analyzed, performing frequency (smoothed amplitude Fourier spectra) and time-frequency analyses (using Stockwell Transform [28] and band-variable filter [29]) considering the relative floor accelerations.

In order to analyze the accelerometric data referred to the shaking table tests performed using the Rnd and Norcia inputs (see Table 1), all records were first modified using the following processes: Baseline correction;Trends removal;0.1–25 Hz band-pass filter;Smoothing with a Tukey window.

All data refer to the accelerations recorded by the sensors, as described in Figure 5, columns 1 and 3. The relative accelerations were evaluated considering as reference stations those located at the base of the model for each horizontal direction. Then, considering only the random tests, the amplitude spectra and related modulus were evaluated using the following relationship:(3)$$\left|A_{r}\left(\omega \right)\right|=\left|\int_{-\infty }^{+\infty }a_{r}\left(t\right)\cdot e^{-i\omega t}dt\right|$$
where $a_{r}$ represents the relative floor acceleration. The modulus of the relative acceleration of spectra $\left|A_{r}\left(\omega \right)\right|$ were smoothed using a 75-point moving average filter implemented in MATLAB^®^. Figure 16 shows the smoothed amplitude Fourier spectra evaluated in the experimental model, considering the data acquired during the NRC_01g test. The same results were also used to evaluate the mode shapes associated to the first fundamental modal frequencies related to the x- and y-directions.

It is worth noting that the spectral characteristics evaluated using this kind of approach are representative of the stationary behavior of the model. When damage processes are under way, the structural modal frequencies change over time during a strong motion excitation; then, the linear modal parameters may not be fully representative of the actual state of damage of the structure. Therefore, structural damage time-frequency analyses allow a more proper evaluation of the actual state of damage, because they are able to provide the frequency changes over time.

With the aim of evaluating the minimum value of the modal frequencies exhibited by the structure during the nonstationary phase, ST analyses were performed on each of the top floor accelerometric recordings (along both the x- and y-directions). The ST was evaluated using the following relationship [28]:(4)$$A_{r}\left(\tau ,f\right)=\frac{\left|f\right|}{2\pi }\int_{-\infty }^{+\infty }a_{r}(t)\cdot e^{-\frac{{\left(\tau -t\right)}^{2}\cdot f^{2}}{2}}\cdot e^{-i\cdot 2\cdot \pi \cdot f\cdot t}dt$$
where *t* is the time; *f* is the frequency; and *τ* is a parameter controlling the position of the Gaussian window along the time axis. An example of the application of the ST is presented in Figure 17, and it is related to the test where the NRC_0.3g earthquake was applied.

The ST analysis allowed for the evaluation of the fundamental frequency variation of the structure during the strong motion test. Along the x-direction, the minimum value of the fundamental frequency exhibited by the structure during the NRC_0.3g test was equal to 1.86 Hz. Similar results were found using STIRF [30] analyses on the same accelerometric recording, not shown here for the sake of brevity. More details concerning the nonlinear behavior of the structure exhibited during the shaking table tests are reported in Section 6.

## 6. Results

After the shaking table tests, a visual inspection was carried out in order to assess the crack pattern and state of damage to the structure. The visible cracks were essentially concentrated at the beam column joints on first floor (Figure 18). Fewer cracks were also visible at the beam column joints on second floor (Figure 19). Only a limited number of small cracks arose at the column’s base.

The surveyed crack positions, orientation, size, and openings were considered in order to assess the damage grade of the structure. Initially, the damage grade was detected according to the “Agibilita’ e Danno nell’Emergenza Sismica (AeDES)” (“Building Operability and Damage during the Post–Earthquake Emergency”) survey form, which is used by the Italian Civil Protection for assessing the usability and damage of buildings after earthquakes [41]. Then, the damage was also assessed with the European Macroseismic Scale of 1998 (EMS98) [42] using the damage grades conversion table proposed in [43]. The damage that resulted was between D3 and D4 for the vertical members (columns) and D1 for the horizontal members (i.e., beams and floors).

The distance between the 3D MCS markers was monitored to assess the crack openings and widths. Table 3 summarizes the distribution of the cracks and their widths after the NRC_100 shaking table tests. An analysis of the markers’ distances confirmed significantly that the distribution and severity of the cracks were concentrated at the first floor beam column joints. In particular, the number of cracks on the first floor with a width greater than 1 mm was more than double that detected on the second floor.

The inter-story drift calculated using the markers during the shaking table sequence is reported in Figure 20. The first and second floor drifts were very similar for the PGA lower than 0.4 g. Then, the second floor drift in the x-direction resulted in the largest until the final seismic test. The second floor drift was substantial because of the behavior of the base of the second floor columns, which constituted the upper part of the first floor beam column joints.

The 3D laser scanning survey after the seismic tests confirmed the deformation of the structure concentrated at the beam column joints, causing substantial rotation of the columns, while the floor slabs did not exhibit relevant deformation and remained essentially undamaged.

In Figure 21a, a comparison of the f_r_ estimates in the Rnd tests with the accelerometers, markers, and MM videos is depicted vs. the PGA of the previous shaking table test. In order to assess the state of damage of the structure, a damage index (D) can be formulated based on f_r_, according to the following equation:(5)$$D=100\left[1-{\left(\frac{f_{r}}{f_{0}}\right)}^{2}\right]$$
where f_0_ is the first modal frequency in Rnd_01, which was performed before the seismic tests so that it was representative of the undamaged structure. Figure 21b shows the estimated D values.

Table 4 summarizes the estimated values of f_r_ and the corresponding errors with reference to the frequencies obtained by the accelerometers, assumed as the most conventional and consolidated method. The markers of the used 3D MCS reported errors from –0.10 to +0.08 Hz, with an RMS error of 0.06 Hz. The MM–based estimates provided errors between −0.38 and +0.11 Hz, with an RMS error of 0.16 Hz, with a higher camera speed (150 fps) but LD camera resolution. When using the HD mode resolution and lower speed (50 fps), the error in the MM estimate of f_r_ was in the range −0.08 to −0.22 Hz, with an RMS error of 0.07 Hz. It is interesting to note that the MM result was at 50 fps, which was more representative of the CGCs, slightly better than at a higher speed (150 fps), indicating that when the studied frequencies do not require high-speed acquisitions, the pixel resolution seems to play a more relevant role.

In addition, MM slightly underestimated the f_r_ identified using accelerometers, providing negative average error values (–0.08 Hz and –0.15 Hz with the LD and HD modes, respectively). This is probably because of the MM algorithm, which affects the correct timing of the motions.

The first modal shape obtained by the MM videos and 3D MCS markers are illustrated in Figure 22. Here, the modal shape during the Rnd_01 (undamaged structure) and Rnd_12 (final damaged condition) tests can be compared. The modal shape was characterized by markers at only three measurement positions (base, first floor, and second floor), because the marker locations were mainly concentrated at the beam column joints to monitor the cracking where most of the damage was expected. Of course, many more markers could be used on the columns. However, the number of usable markers was limited in comparison to the positions detectable by video pixels. In fact, the modal shape by the MM video was obtained by digitizing 100 points, meaning one point every 4 cm in height. In Figure 22a, the effect of the beam column joints is extremely evident, especially in terms of the relationship between the flexural stiffness of the beams and columns.

As it is often the case in low-rise and mid-rise buildings, the relative stiffness of frame members lies in between flexural-type and shear-type behavior. As a consequence, both beams and columns bend in a double curvature, which typically results in an almost shear-type response. Moreover, the rotational flexibility of the columns at the base of the building is determined by the degree of fixity at the structure’s base. The slightly negative rotation detected at the structure’s base and both beam column joints of the first and second floors can partially be due to the nonperfectly vertical positioning of the MM camera. However, such rotations were on the order of 0.1 mm in 50 cm, which is very small. In Figure 22b, it is interesting to note that the double curvature of the first floor beam column joint disappeared completely in the Rnd_12 test, which is the effect of advanced damage to the joint, making the column unable to effectively connect to the first floor beam. The curvature at structure’s base and the second floor are still visible but clearly reduced with respect to the initial undamaged condition as in Rnd_01, which indicates less damage.

In Figure 23a, an analysis of the evolution of the first modal shape is illustrated in terms of the curvature difference between Rnd_01 and Rnd_12. As a confirmation, the curvature difference using MM showed higher values in correspondence with the first floor beam column joint, locating with remarkable accuracy the main damage.

The location of the damage seemed to be even more accurately indicated by the zero crossings of the curvature difference variation in Figure 23b, even if zero crossings cannot provide an estimate of the severity of the damage.

In order to prove the consistency of the obtained results, the modal shape was extracted using the three amplest oscillations (Figure 24a). The corresponding curvature differences depicted in Figure 24b clearly confirm the damage location obtained with the absolute maximum FMD and minimum FZD frames.

Using the modulus of the Fourier spectra of the relative accelerometric Rnd recordings acquired from the top floor, the main modal frequencies were evaluated along both the x- and y-directions. Using the NRC accelerometric recordings, time-frequency ST analyses were performed with the aim of evaluating the minimum value of the fundamental frequency exhibited by the structure during each shaking table test. In Figure 25, a comparison of the estimated modal frequencies along the x- and y-directions is shown.

It is worth noting that the higher the level of structural damage, the greater the difference between the structural natural frequencies estimated using the random tests (stationary behavior) and those evaluated using the Stockwell transform during the strong motion phase.

Considering the modal frequencies values depicted in Figure 25, in Figure 26 the differences are quantified as a function of the nominal PGA.

Analyzing the results shown in Figure 25 and Figure 26, the importance of using time-frequency analysis techniques is demonstrated, as the correct evaluation of the structural modal frequencies of the damaged structure and, consequently, the correctly estimation of the relative structural damage can be observed. Using the performed random tests, the fundamental mode shape evolution relative to the x-direction was evaluated and depicted in Figure 27.

The curvature variation was evaluated on the interpolated mode shapes retrieved using piecewise cubic Hermite interpolating polynomials, where the most damaged floor was the first floor. The results obtained for each random test in terms of the curvature variation confirmed what was found in the laboratory using traditional visual inspections and using the MM–based technique.

In order to identify the most damaged floor using the accelerometric dataset, the curvature variations were evaluated using the procedure described in [29,44,45] and shown in Figure 28.

## 7. Conclusions

The present work proposed an MM–based procedure to assess the state of damage of a framed RC building using low-cost video-based equipment and processing. The proposed procedure was validated through shaking table experiments, reproducing the damage to an Italian building typology subjected to a recent earthquake. In particular, the MM–based procedure aimed at extracting the most relevant modal parameters (frequencies and shapes) for the structural health monitoring of buildings.

The results obtained using the proposed procedure were compared for validation with utilizing methods that were more conventional and consolidated measurement systems, including accelerometers, a 3D laser scanner, and an optical 3D MCS.

The advanced analyses of the accelerometers data focused on the detection of structural damage using both natural frequencies and mode shape variations. The former were evaluated using Stockwell transform of NRC accelerometric recordings. Differences greater than 40% with respect to the natural frequencies obtained using Rnd recordings were evaluated, at least for the most damaged conditions. Mode shape variations, on the other hand, were evaluated using Rnd data, calculating the curvature parameters on interpolated mode shapes recovered using piecewise cubic Hermite interpolating polynomial. The results obtained for each random test in terms of the curvature variation located the most damage from the first floor.

Moreover, the 3D laser scanning surveys and optical 3D MCS marker displacements confirmed the location of the main damage on the first floor.

The final 3D laser survey detected the structure’s main deformations at the beam column joints, while the floor slabs substantially lacked any deformation. In addition, the 3D MCS markers’ relative displacement revealed that the number and width of the main cracks were concentrated at the first floor’s beam column joints. Moreover, through the relative displacements of the markers, the structural damage was also evaluated through the evolution of the inter-story drifts in the seismic tests. The modal frequencies of the marker data were also calculated and were in agreement with the accelerometers data.

As for the estimates of the modal frequency, the proposed MM–based procedure provided results with reasonable accuracy and repeatability with respect to the results obtained analyzing the accelerometric recordings in stationary conditions. The MM–based estimates had quite similar errors with either higher camera speed (150 fps) but an LD camera resolution and using an HD mode resolution and a low speed (50 fps). This latter acquisition set-up configuration, which can be considered more representative of CGCs typical camera parameters, as well as low cost, providing even slightly better accuracy than acquisition at higher speed (150 fps), proving that already reasonably affordable cameras have speed levels appropriate for detecting common building frequencies.

The results in terms of the detection and analysis of the modal shape curvature after seismic shocks was probably the most relevant and interesting, because of the potential to locate damage to structures. The MM-based procedure accurately located the main structural damage. It was detected at the first floor’s beam column joints, and it was found even by traditional visual inspections, as well as in remarkable accordance with advanced methods of analysis of accelerometric data: 3D laser surveys and 3D MCS markers. Moreover, mode shapes could be obtained using image analysis with greater definition (given its raster nature) than with the use of sparse optical sensors placed on the structure, which makes the video-based procedure a method with high potential even for more complex structural typologies than just framed building.

Therefore, the main novelty of this study was highlighting a simple and low-cost procedure based on the MM method to estimate with good accuracy the intensity and location of damage to a multistory RC framed prototype, which is highly representative of existing buildings in Italy. Even if methods based on conventional contact sensors are still much more accurate and reliable, as well as providing the possibility to analyze the structural behavior during a strong motion phase (i.e., nonlinear conditions), the presented video-based method shows the potential to provide a valuable preliminary estimate of the state and location of damage, almost in stationary conditions, which can help in the management and planning of further extensive monitoring campaigns. Of course, the proposed procedure is also applicable to other construction typologies, such as masonry and steel. The main limitations of this paper are related to the frequency range of interest and the amplitude of the ambient vibration of the structure, such that the equipment’s camera speed and resolution should be appropriate. Future steps and improvements are essentially related to the main challenges of these kinds of techniques: (i) the present study was conducted through a laboratory experiment with an almost real-scale physical model, but studies conducted in the field still present more complicated site conditions, such as the presence of nonstructural components; (ii) the quality of the images captured in the field are significantly affected by the surrounding environmental conditions, such as lighting stability, edge shelter by rain, snow and fog, ground vibration, etc.; and (iii) the further development of more effective algorithms to improve the reliability of the results affected by the abovementioned disturbances are currently underway and may take advantage of recent advances in artificial intelligence and other innovative processing techniques.

## Figures and Tables

**Figure 1 sensors-23-05303-f001:**
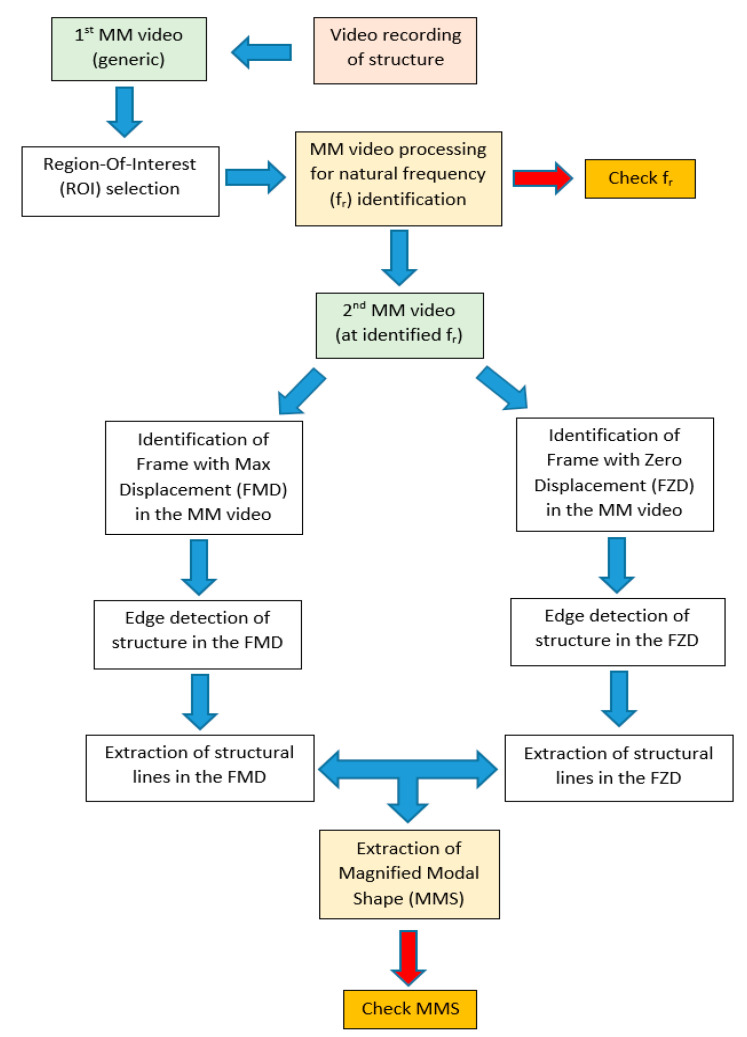
Flowchart of proposed video-based MM processing and analysis method.

**Figure 2 sensors-23-05303-f002:**
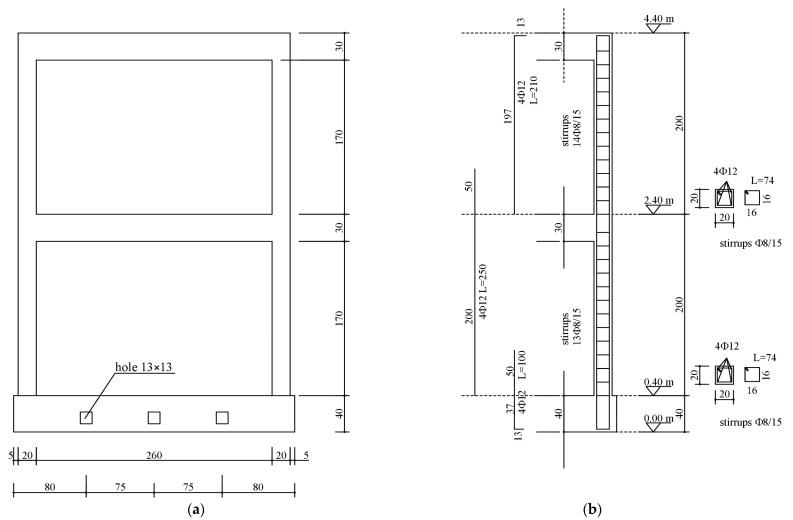
Tested structure: (**a**) building’s dimensions (cm); (**b**) steel reinforcement of columns.

**Figure 3 sensors-23-05303-f003:**
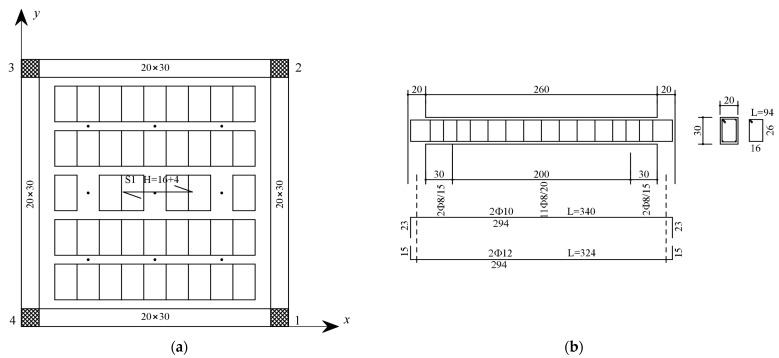
Details of the tested structure’s floor: (**a**) top view of the slabs; (**b**) steel reinforcement of the beam.

**Figure 4 sensors-23-05303-f004:**
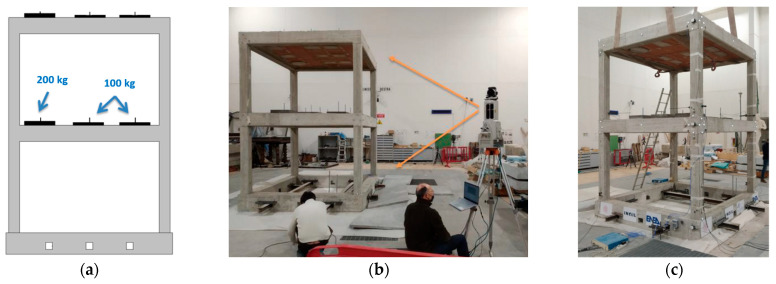
Test set-up: (**a**) additional load mass and geometry; (**b**) 3D geometry survey by laser scanner; (**c**) final set-up ready for shaking table tests.

**Figure 5 sensors-23-05303-f005:**
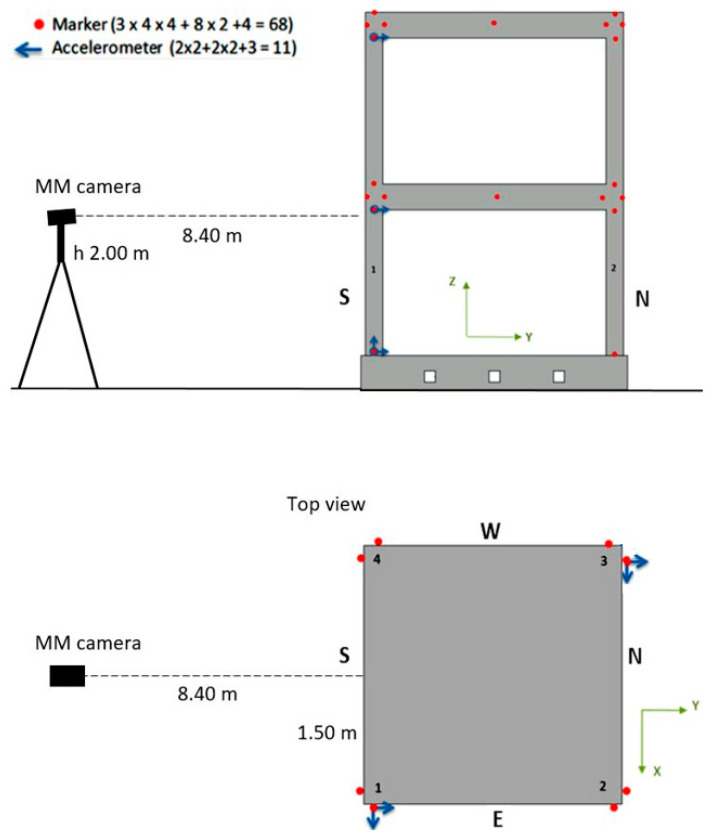
Instrumentation positions in the lateral (**up**) and top (**bottom**) views. Columns are numbered from 1 to 4 counterclockwise. Building facades are named S, E, N, and W according to cardinal points.

**Figure 6 sensors-23-05303-f006:**
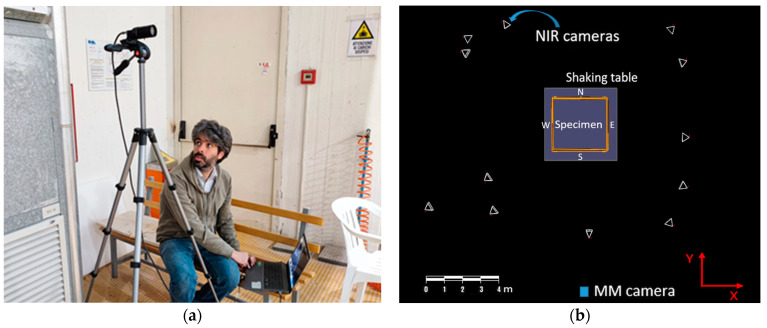
Instrumentation set-up: (**a**) MM camera acquisition station; (**b**) top view of the acquisition geometry of the optical 3D MCS NIR cameras (white triangles).

**Figure 7 sensors-23-05303-f007:**
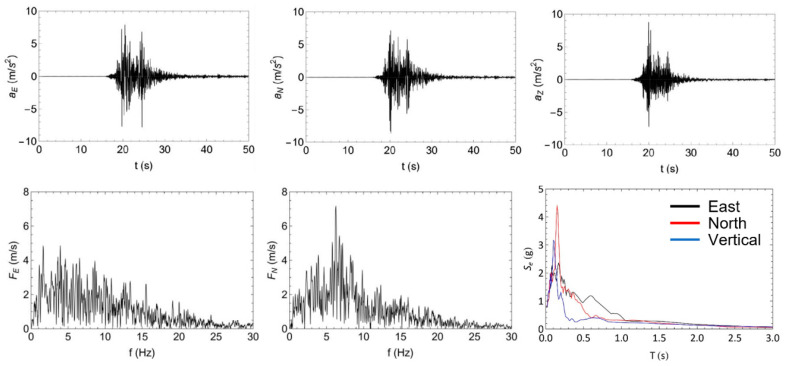
Characterization of the reference input accelerograms of the seismic NRC shaking table tests (record NRC station, 30 October 2016, 06:40 UTC, Norcia, Italy).

**Figure 8 sensors-23-05303-f008:**
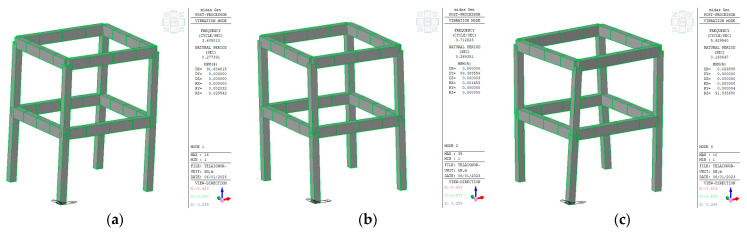
First three modal shapes: (**a**) first bending mode in the *x*-direction; (**b**) first bending mode in the *y*-direction; (**c**) first torsional mode.

**Figure 9 sensors-23-05303-f009:**
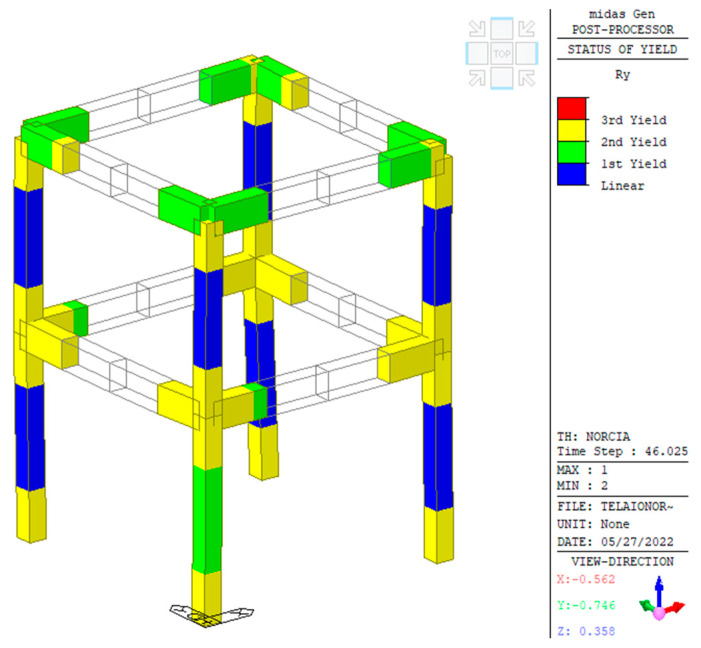
Yielding state at the end of the analysis with unscaled seismic input (NRC_100).

**Figure 10 sensors-23-05303-f010:**
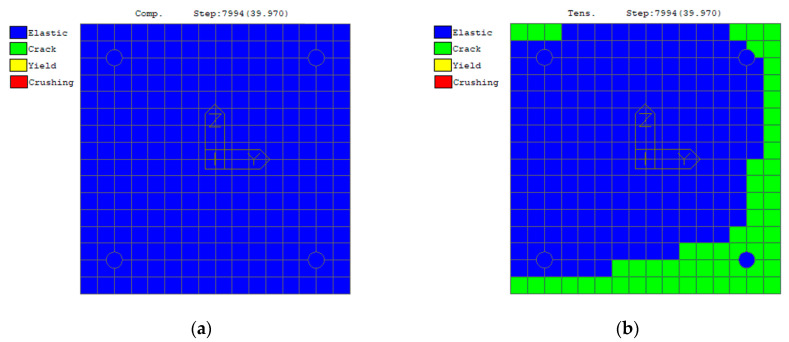
Analysis of a column’s top at a seismic input with a scale factor of 0.1: (**a**) tension; (**b**) compression.

**Figure 11 sensors-23-05303-f011:**
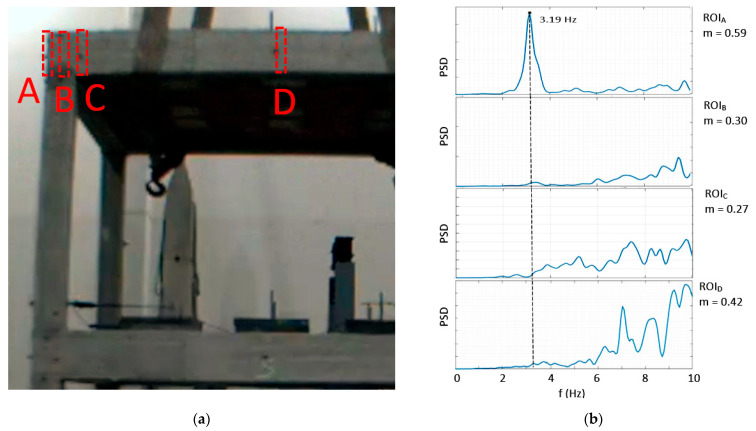
Comparison of ROIs with different image contrasts: (**a**) details of the selected ROIs (red, dotted lines), named A to D; (**b**) PSD and contrast index *m* for each selected ROI.

**Figure 12 sensors-23-05303-f012:**
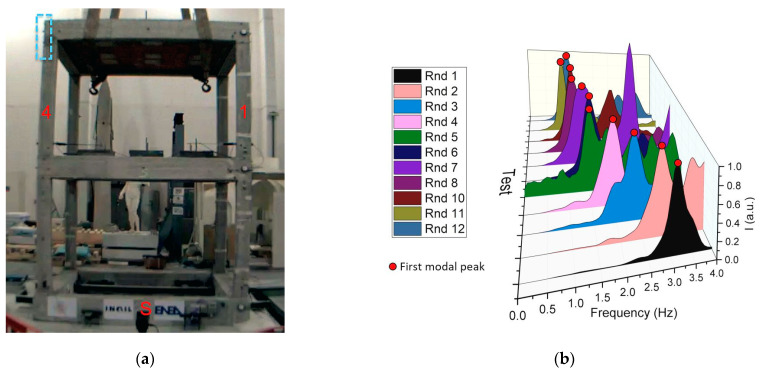
Video processing using motion magnification (MM): (**a**) final ROI (blue, dotted line) at the south (S) side of the top column 4; (**b**) identification of the first mode peaks in the frequency domain for all white noise Rnd shaking table tests.

**Figure 13 sensors-23-05303-f013:**
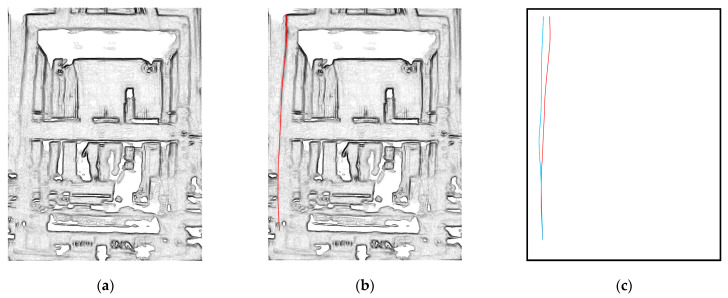
Video processing for the modal shape extraction: (**a**) FMD after edge detection; (**b**) identification of the structural line of column 4 (red line) in the FMD; (**c**) extracted structural line of column 4 from the FMD (red line) and from the FZD (blue line).

**Figure 14 sensors-23-05303-f014:**
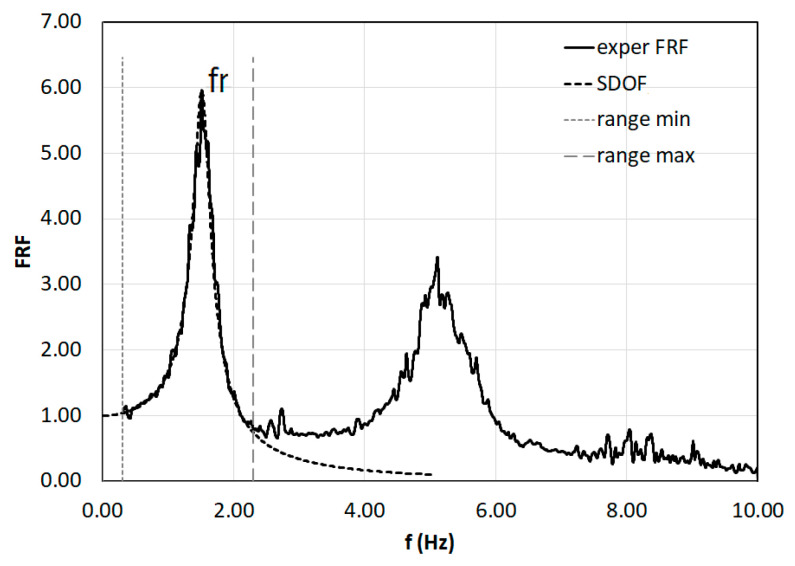
Identification of the first modal frequency (f_r_) of the FRF of the marker data (exper FRF).

**Figure 15 sensors-23-05303-f015:**
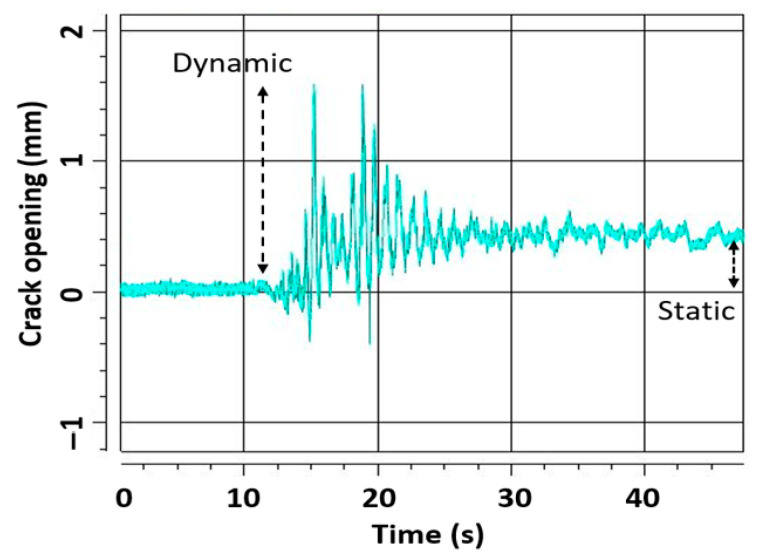
Typical crack opening during a seismic test. The shake began at 11 s.

**Figure 16 sensors-23-05303-f016:**
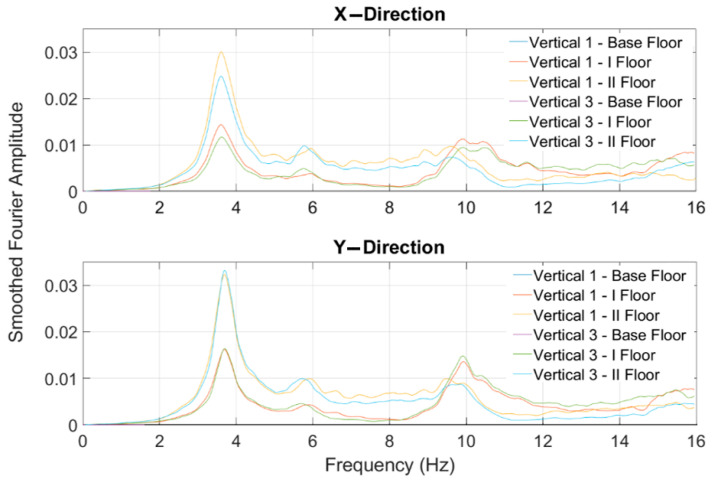
Smoothed amplitude Fourier spectra evaluated for the RND_01 test.

**Figure 17 sensors-23-05303-f017:**
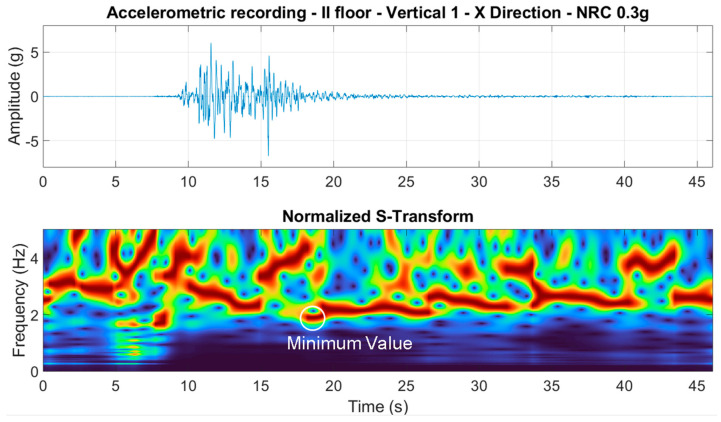
Stockwell transform (bottom graph) evaluating the NRC_0.3g test (top graph), related to column 1, second floor, in the x-direction. In the bottom graph, false colors indicate values from 0 (blue) to maximum (red).

**Figure 18 sensors-23-05303-f018:**
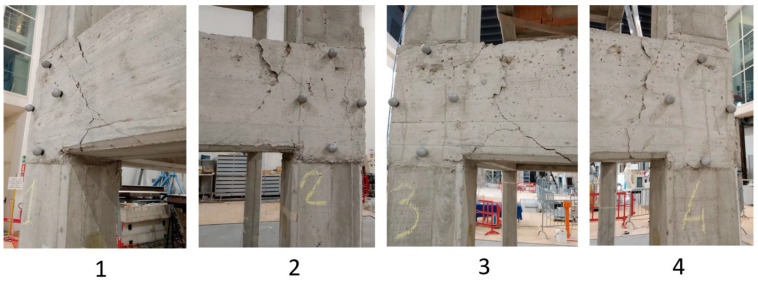
Details of the damage at beam column joints on first floor (the numbers indicate the columns).

**Figure 19 sensors-23-05303-f019:**
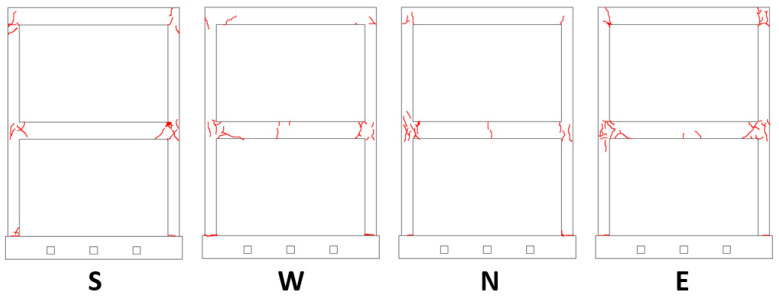
Cracks (red lines) at the end of the seismic tests. Letters indicate the facades.

**Figure 20 sensors-23-05303-f020:**
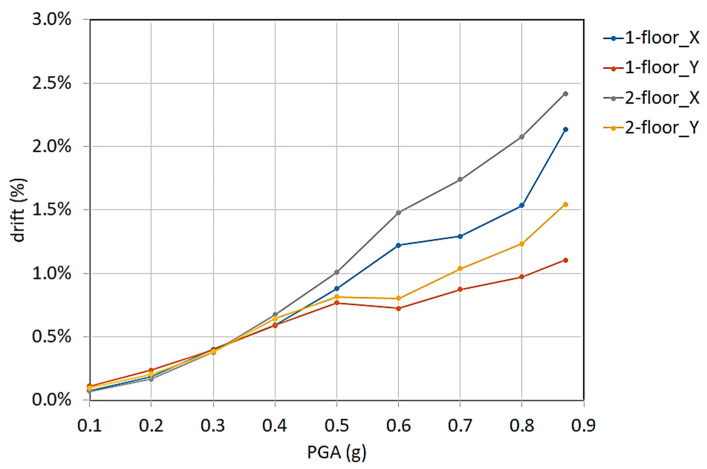
Inter-story drifts (1 and 2 floors) in the x- and y-directions vs. shaking table PGA.

**Figure 21 sensors-23-05303-f021:**
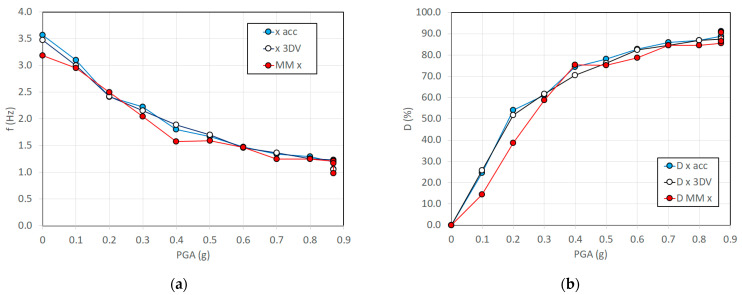
First modal frequency (f_r_) of the tested structure calculated with marker data (3DV), accelerometers (acc), and motion magnification (MM) in the x-direction: (**a**) evolution of f_r_ in white noise random tests during shaking table sequence; (**b**) corresponding damage index D%.

**Figure 22 sensors-23-05303-f022:**
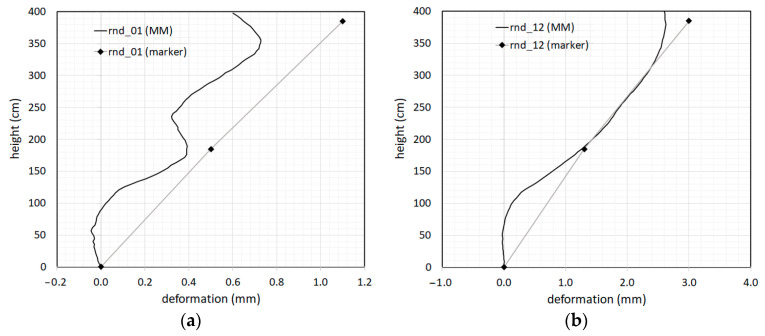
First modal shape calculated using the MM and markers in the *x*-direction: (**a**) modal shape extracted with the “Rnd_01” test; (**b**) modal shape extracted with the “Rnd_12” test.

**Figure 23 sensors-23-05303-f023:**
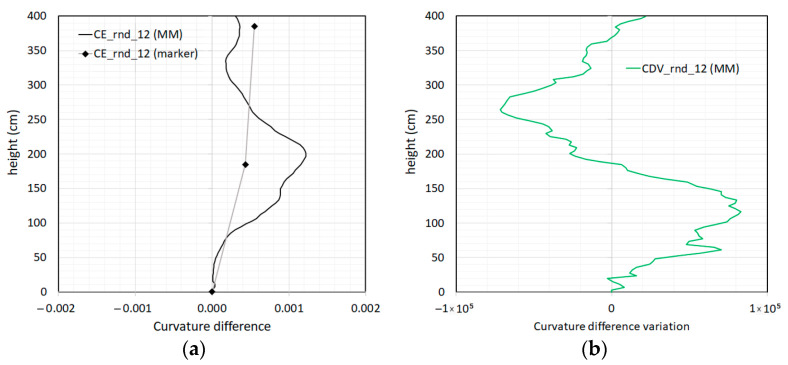
Analysis of the first modal shapes: (**a**) modal shape curvature difference (CE) between the “Rnd_12” and “Rnd_01” tests; (**b**) corresponding curvature difference variation (CDV).

**Figure 24 sensors-23-05303-f024:**
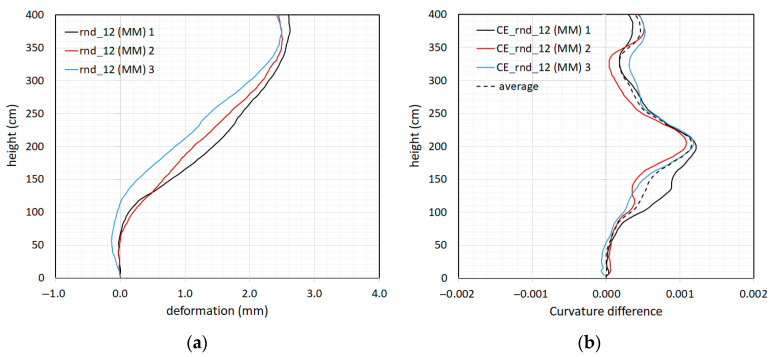
Repetition of the modal shape extraction using the three amplest oscillations (“1”, “2”, and “3”)**:** (**a**) modal shape extracted using the “Rnd_12” test; (**b**) modal shape curvature difference (CE) between the “Rnd_12” and “Rnd_01” tests.

**Figure 25 sensors-23-05303-f025:**
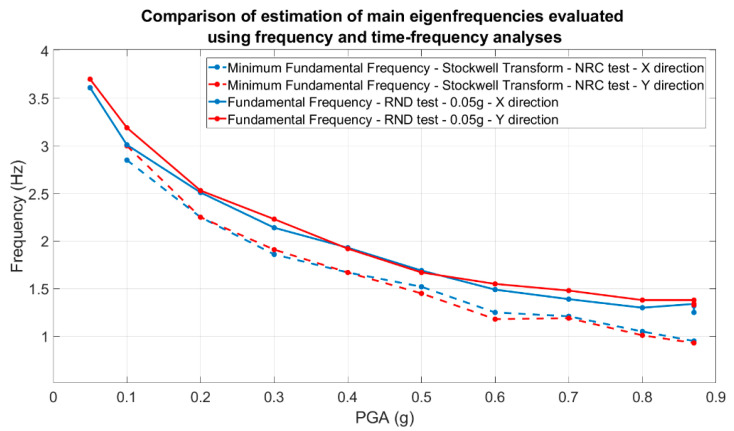
Comparison of the main structural modal frequencies evaluated using random tests (stationary values of the modal frequencies) and Stockwell transform (minimum value of the modal frequencies).

**Figure 26 sensors-23-05303-f026:**
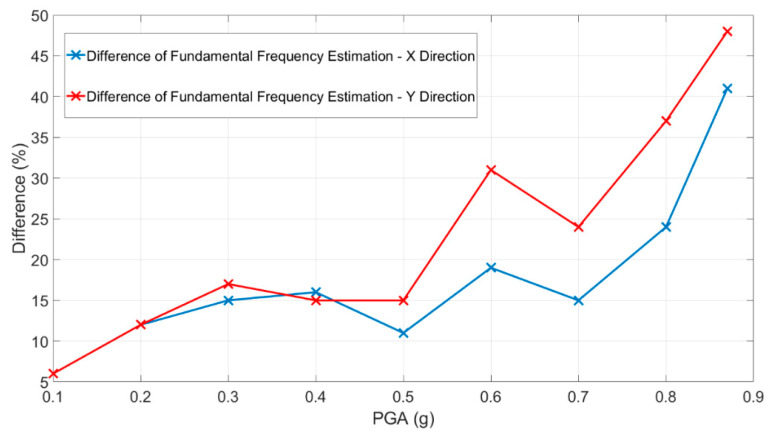
Differences in the estimation of the structural modal frequencies using random tests (stationary values in the modal frequencies) and Stockwell transform (minimum value of the modal frequencies).

**Figure 27 sensors-23-05303-f027:**
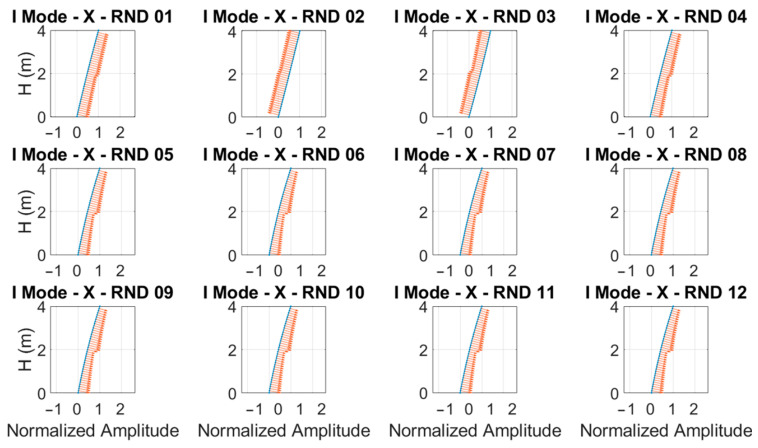
Fundamental mode shape (blue lines) and related curvature vectors (red arrows) evaluated along the *x*-direction using random tests.

**Figure 28 sensors-23-05303-f028:**
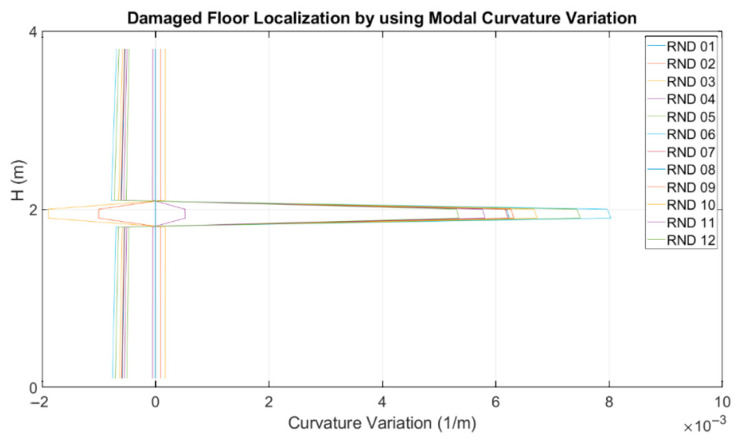
Damage level localization by using the modal curvature variation.

**Table 1 sensors-23-05303-t001:** Shaking table testing sequence.

Test	Test Type	Nominal PGA (g)	Notes
Rnd_01	White noise triaxial	0.05	Dynamic identification test, 2 min duration
NRC_01g	Seismic triaxial	0.10	11% PGA scaled 30 October 2016 shake, Norcia, Italy
Rnd_02	White noise triaxial	0.05	Dynamic identification test, 2 min duration
NRC_02g	Seismic triaxial	0.20	23% PGA scaled 30 October 2016 shake, Norcia, Italy
Rnd_03	White noise triaxial	0.05	Dynamic identification test, 2 min duration
NRC_03g	Seismic triaxial	0.30	34% PGA scaled 30 October 2016 shake, Norcia, Italy
Rnd_04	White noise triaxial	0.05	Dynamic identification test, 2 min duration
NRC_04g	Seismic triaxial	0.40	45% PGA scaled 30 October 2016 shake, Norcia, Italy
Rnd_05	White noise triaxial	0.05	Dynamic identification test, 2 min duration
NRC_05g	Seismic triaxial	0.50	57% PGA scaled 30 October 2016 shake, Norcia, Italy
Rnd_06	White noise triaxial	0.05	Dynamic identification test, 2 min duration
NRC_06g	Seismic triaxial	0.60	69% PGA scaled 30 October 2016 shake, Norcia, Italy
Rnd_07	White noise triaxial	0.05	Dynamic identification test, 2 min duration
NRC_07g	Seismic triaxial	0.70	80% PGA scaled 30 October 2016 shake, Norcia, Italy
Rnd_08	White noise triaxial	0.05	Dynamic identification test, 2 min duration
NRC_08g	Seismic triaxial	0.80	91% PGA scaled 30 October 2016 shake, Norcia, Italy
Rnd_09	White noise triaxial	0.05	Dynamic identification test, 2 min duration
NRC_100	Seismic triaxial	0.87	Full 30 October 2016 shake, Norcia, Italy
Rnd_10	White noise triaxial	0.05	Dynamic identification test, 2 min duration
Rnd_11	White noise triaxial	0.05	Dynamic identification test, 2 min duration
NRC_100b *	Seismic triaxial	0.87	Full 30 October 2016 shake, Norcia, Italy
Rnd_12	White noise triaxial	0.05	Dynamic identification test, 2 min duration

* Repetition of NRC_100.

**Table 2 sensors-23-05303-t002:** First three modal frequencies using the linear model with the initial and cracked stiffness.

Mode	Frequency (Hz) with Initial Stiffness	Frequency (Hz) with Cracked Stiffness	Mode Type
1	5.125	3.605	Bending in the x-direction
2	5.127	3.713	Bending in the y-direction
3	8.101	5.930	Torsional

**Table 3 sensors-23-05303-t003:** Crack openings and widths under dynamic conditions during the NRC_100 test.

Crack Location	Sum of Crack Widths (mm) ^1^	Number of Cracks with Width >1 mm ^1^	Widest Crack Openings (mm) ^1^
Floor 1	88.2	39	9.9
Floor 2	54.1	19	3.4

^1^ Calculated using 3D MCS marker distances.

**Table 4 sensors-23-05303-t004:** Estimated f_r_ values and errors with MCS marker data (3DV) and motion magnification (MM) with respect to the accelerometer data (acc).

	Estimated f_r_ (Hz)			Error (Hz)		MM Camera Parameters	
Test	acc	3DV	MM	3DV	MM	fps	Resolution (px)
Rnd_01	3.57	3.48	3.19	−0.09	−0.38	150	632 × 508
Rnd_02	3.10	3.00	2.95	−0.10	−0.15	50	1264 × 1016
Rnd_03	2.42	2.42	2.50	0.00	0.08	150	632 × 508
Rnd_04	2.23	2.15	2.05	−0.07	−0.18	150	632 × 508
Rnd_05	1.81	1.89	1.58	0.08	−0.23	150	632 × 508
Rnd_06	1.67	1.70	1.59	0.03	−0.08	50	1264 × 1016
Rnd_07	1.48	1.46	1.47	−0.02	−0.01	150	632 × 508
Rnd_08	1.34	1.37	1.25	0.03	−0.09	150	632 × 508
Rnd_09	1.29	1.26	1.25	−0.03	−0.04	150	632 × 508
Rnd_10	1.19	1.23	1.21	0.04	0.02	150	632 × 508
Rnd_11	1.20	1.22	0.98	0.02	−0.22	50	1264 × 1016
Rnd_12	1.07	1.06	1.17	−0.01	0.10	150	632 × 508

## Data Availability

Not applicable.

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
