# Peer review of "Advanced Video-Based Processing for Low-Cost Damage Assessment of Buildings under Seismic Loading in Shaking Table Tests"

_sensors, 2023, doi:10.3390/s23115303_

Round 1

Reviewer 1 Report

This paper presents a comprehensive evaluation of a low-cost video-based method's efficacy in assessing structures subjected to seismic loading. The introduction clearly highlights the research motivation, drawing attention to the affordability and adaptability of the method in comparison to traditional techniques. The paper's main concern revolves around the method's accuracy and reliability, which the study aims to validate. The research design is thoughtful with both simulations and experiments, and the analysis is logical and critical. Discussions and visualizations are also well performed.

To enhance the quality of the paper, the following suggestions could be considered:

A more exhaustive review of video-based methods could be beneficial. This could include an exploration of applications where Unmanned Aerial Vehicles (UAVs) have been employed.

The Motion Magnification (MM) method relies on two frames from a video – one depicting maximum displacement and the other minimum. However, it might be useful to examine additional frames to confirm that the chosen maximum and minimum displacements are not outliers, which could be potentially caused by camera shaking for example.

The Region-Of-Interest was chosen with consideration for both the structurally critical region and the image's high contrast region. However, in practical situations, these two regions may not necessarily overlap. It would be valuable if the authors could employ the presented experiments to demonstrate how MM performs if a low contrast region is selected.

And one typo: caption of Figure 8 b should be 'y-direction'.

Author Response

Reply to the reviewers’ comments

The reviewers and the editor are thanked for their efforts to review the manuscript and for their constructive comments. These have helped to improve the quality of the paper. Detailed responses to the actions taken to address these comments are provided below.

Reviewer 1:

This paper presents a comprehensive evaluation of a low-cost video-based method's efficacy in assessing structures subjected to seismic loading. The introduction clearly highlights the research motivation, drawing attention to the affordability and adaptability of the method in comparison to traditional techniques. The paper's main concern revolves around the method's accuracy and reliability, which the study aims to validate. The research design is thoughtful with both simulations and experiments, and the analysis is logical and critical. Discussions and visualizations are also well performed.

To enhance the quality of the paper, the following suggestions could be considered:

  1. A more exhaustive review of video-based methods could be beneficial. This could include an exploration of applications where Unmanned Aerial Vehicles (UAVs) have been employed.

Addressing comment # 1: The authors thank the reviewer for this suggestion that helps completeness of introduction. A brief review of video-based applications where Unmanned Aerial Vehicles (UAVs) are employed was added to the introduction as follows: ‘Video-based methods include applications where Unmanned Aerial Vehicles (UAVs) are employed [15-16]. UAV methods for damage detection and quantification are mainly focused on high-resolution visual inspection of structures that allow accurate cracking and defects survey. Also, UAV-based approaches were explored for structural  identification, but they need the development of effective algorithms to compensate for the effect of the UAV’s egomotion in the displacement signals, which is certainly a remarkable complication respect to the use of fixed cameras.’ (lines 70-77). Related following references were added:

  • 15 - Molina, A.A.; Huang, Y.; Jiang, Y. A Review of Unmanned Aerial Vehicle Applications in Construction Management: 2016–2021. Standards 2023, 3, 95–109.
  • 16 - Kapoor, M., Katsanos, E., Nalpantidis, L., Winkler, J., & Thöns, S. (2021). Structural Health Monitoring and Management with Unmanned Aerial Vehicles: Review and Potentials. Technical University of Denmark, Department of Civil Engineering.

  1. The Motion Magnification (MM) method relies on two frames from a video – one depicting maximum displacement and the other minimum. However, it might be useful to examine additional frames to confirm that the chosen maximum and minimum displacements are not outliers, which could be potentially caused by camera shaking for example.

Addressing comment # 2: The authors thank the reviewer for this suggestion that makes the study more solid, improving the paper. We added to the text as follows: “In order to verify the consistency of the results the proposed method can be analogously applied to more couples of frames corresponding to relative maximum and mini-mum displacements in any period of oscillation of the studied structure in the magnified video at a given modal frequency. For example, in a magnified video with duration of 20 s at the modal frequency of 3 Hz (e.g. three oscillation per second) we have 60 oscillations, so that we can identify 60 couples of relative FMD and FZD frames. Finally, an averaged modal shape can be calculated and analyzed. However, this would imply much greater computational time. Consequently, we limited the extraction of the modal shape to the three amplest oscillations just to demonstrate that the result obtained with only the absolute maximum FMD and minimum FZD frames is not an outliner”.(Lines 210-220).

The corresponding results are shown in new figure 24 and discussed in lines 596-620 as follows: “As it is often the case in low-rise and mid-rise buildings, the relative stiffness of frame members lies in between the flexural-type and the shear-type behavior. As a consequence, both beams and columns bend in double curvature, which typically gives an almost shear type response. Moreover, the rotational flexibility of columns at the base of the building is determined by the degree of fixity at structure base. The slight negative rotation detected at structure base and both beam-column joints of first and second floors can partially due to non-perfectly vertical positioning of the MM camera. However, such rotations are in the order of 0.1 mm in 50 cm, which is very small. In Figure 22b it is interesting noting that the double curvature at first floor beam-column joint disappears completely in the test Rnd_12, which is the effect of advanced damage of the joint that is unable to efficiently connect columns and first floor beam. Curvature at structure base and second floor are still visible, but clearly reduced with respect to initial undamaged condition in Rnd_01, indicating more limited damage. In Figure 23a the analysis of the evolution of first modal shape is illustrated in terms of curvature difference between Rnd_01 and Rnd_12. As a confirmation, curvature difference by MM shows higher values in correspondence of the first floor beam-column joint, locating with remarkable accuracy the main damage. Location of damage seems even more accurately indicated by the zero crossings of the curvature difference variation in Figure 23b, even if zero crossings cannot provide an estimate of damage severity. In order to prove the consistency of the obtained result, the modal shape was extracted using the three amplest oscillations (Figure 24a). The corresponding curvature differences depicted in Figure 24b clearly confirmed the damage location obtained with the absolute maximum FMD and minimum FZD frames.”

  1. The Region-Of-Interest was chosen with consideration for both the structurally critical region and the image's high contrast region. However, in practical situations, these two regions may not necessarily overlap. It would be valuable if the authors could employ the presented experiments to demonstrate how MM performs if a low contrast region is selected.

Addressing comment # 3: The authors thank the reviewer for the interesting observation. We added a comparison between 4 different ROIs with same size and equivalent positions in terms of modal shape deformation, but with different image contrast. In the text we added as follows: “…four tentative ROIs, named with subscript from A to D, were selected at the upper floor, where the simulated first bending mode in x-direction provides the highest values of deformation (Figure 11a). The selected ROIs were assessed in terms of image contrast by calculating the Michelson index m, which is defined as follows [38]:

(1)

where Imax and Imin are the highest and lowest pixel values in the ROI, respectively. Then the ROIs’ pixels were analyzed in the frequency domain and the PSD was calculated. As illustrated in Figure 11b, ROIA has with the highest m value and provides a good identification of the modal frequency. In fact, contrast in ROIA is determined by the difference in color between the column edge and the white background, while contrast in the other ROIs is very poor, as it is only due to slight heterogeneities in the color of concrete and the presence of markers. Thus, the final ROI was chosen enlarging ROIA, as shown in Figure 12a. The PSD was then calculated for each Rnd test (Figure 12b).” (Line 361-373)

We also added a new Figure 11 and a reference as follows:

  • 38 - Peli, E. Contrast in Complex Images. Journal of the Optical Society of America A. 1990, 7(10), pp. 2032–2040.

  1. And one typo: caption of Figure 8 b should be 'y-direction'.

Addressing comment # 4: The authors thank the reviewer for spotting the typo. The typo was corrected according to reviewer’s suggestion.

Reviewer 2 Report

Dear authors,

Below please find some remarks considering your paper.

·         In the case of the title and the abstract, those elements must be understandable to none familiar with the field reader. It is advised to use no abbreviations if possible. In the abstract, the authors are using abbreviations, which should be avoided in the abstract and those abbreviations should be introduced for the first time in the main text. Otherwise, the abstract is good, however, would ask to emphasise the novelty stronger.

·         Would suggest introducing some more keywords to use for the paper identification

·         Please give the full affiliation- according to the journal template

·         The reference list is good with some minor adjustments to be made in case of missing state-of-the-art elements – look next point.

·         The introduction part is written clearly. However:

o   The introduction is quite short and half of it is the aim, scope and novelty of the authors' solution. The authors in a very limited manner introduce the topic of contact and non-contact measurements and the problem of the need for multiple sensors in case contact methods. The background of the study is not described fully and the authors go straight to video-based methods and motion amplification.

o   No background in seismic monitoring and techniques used. No classification of earthquakes. No pointing out problems like sudden and unexpected high amplitude earthquakes or so-called high activity earthquake swarm events that are potentially damaging to civil engineering structures.

o   No alternative methods are mentioned that are also video-based but deal with the problem of the need for numerous measurement points. Some possibilities for other optical systems that can perform such measurements of even very big objects (buildings) include video recorders and virtual visual sensors (e.g. https://doi.org/10.1177/14759217145228), a digital image correlation (DIC) (e.g https://doi.org/10.1016/j.engstruct.2020.110551) and especially 3D Lase Doppler Vibrometry (e.g DOI: 10.3390/s23031263 ). In this last case, this is an optical system dedicated to vibration and seismic measurements but also possible to measure displacement and is used for quality control also in the case of civil engineering structures or building structural elements. Additionally, the system apart from displacement allows the measurement vibration (natural frequencies, mode shapes and damping) without the need to use accelerometers and another acquisition system. Please make the correction and the beginning of the introduction. Without it, there is no proper background to the research you are presenting.

·         Chapter 2 I good- I would advise however to enlarge Fig.1 – the texts are bearly visible.

·         Chapter 3:

o   Fig 2 and 3 please unify the drawings in case of style, colour etc. Especially Fig2b which Is of a low quality. Apart from that the chapter is very well written and clear with all the information needed.

·         Chapters 4 and 5 are good- no comments on the further improvements.

·         The result presentation is very good. Would only ask, if possible, for a better quality of figures 21 and 22. Maybe change the grey marks, texts etc to black so everything is clearly visible.

·         The conclusion is rather a discussion of the results. Please present stronger the novelty of the paper, application, future steps and improvements.

In conclusion: The paper is clear. The biggest advantage is the presence of both tests in laboratory conditions and simulations. The paper needs however a better presentation of novelty and especially adding some alternative techniques in the introduction and the background of the study.

Due to some small flaws, the reviewer is marking the paper for minor revisions. Hope the authors will use some suggestions to improve this otherwise very interesting and good, in case of results presentation, paper.  

Author Response

Reply to the reviewers’ comments

The reviewers and the editor are thanked for their efforts to review the manuscript and for their constructive comments. These have helped to improve the quality of the paper. Detailed responses to the actions taken to address these comments are provided below.

Reviewer 2:

Dear authors,

Below please find some remarks considering your paper.

  1. In the case of the title and the abstract, those elements must be understandable to none familiar with the field reader. It is advised to use no abbreviations if possible. In the abstract, the authors are using abbreviations, which should be avoided in the abstract and those abbreviations should be introduced for the first time in the main text. Otherwise, the abstract is good, however, would ask to emphasise the novelty stronger.

Addressing comment # 1: The authors thank the reviewer for the overall positive comment and suggestion to improve it. Abbreviations in the abstract were eliminated, according to reviewer’s suggestion. Moreover, the novelty of the paper was more emphasised as follows: “the main novelty of the study is to point out a simple procedure with high potentiality of extracting and analyzing the modal parameters, with special focus on the analysis of the modal shape’s curvature, which provides accurate information on the location of the damage in a structure, while using a non-contact and low-cost method.” (lines 27-30)

  1. Would suggest introducing some more keywords to use for the paper identification

Addressing comment # 2: The authors thank the reviewer for the suggestion. Two keywords were added: vibrational monitoring; modal parameter identification; damage location. (lines 31-32)

  1. Please give the full affiliation- according to the journal template

Addressing comment # 3: The authors thank the reviewer for the indication. The full affiliation of each coauthor was corrected according to the journal template.

  1. The reference list is good with some minor adjustments to be made in case of missing state-of-the-art elements – look next point.

Addressing comment # 4: The authors thank the reviewer for the suggestion. The suggested minor adjustments were made according to the reviewer’s comments in the next points 5, 6 and 7.

The introduction part is written clearly. However:

  1. The introduction is quite short and half of it is the aim, scope and novelty of the authors' solution. The authors in a very limited manner introduce the topic of contact and non-contact measurements and the problem of the need for multiple sensors in case contact methods. The background of the study is not described fully and the authors go straight to video-based methods and motion amplification.

Addressing comment # 5: The authors added to the introduction a more extensive background. A part on the topic of contact and non-contact measurements and the problem of the need for multiple sensors in case contact methods as follows: “Besides, these conventional contact methods usually require periodical maintenance and potential replacement, which causes extra costs with the passing of time. In addition, using many contact sensors adds by definition extra mass loading to the monitored structure, which can have an obvious negative impact on lightweight structures.” (Lines 51-55) “In alternative, non-contact methods can be used, with the advantage of eliminating the challenges of contact-based sensors. Recent advances of technology provide several tools for contactless sensing which is overtaking traditional contact sensing. Among the emergent approaches on non-contact methods and tools, the most relevant techniques are vision-based, wireless, embedded sensors and guided waves [12].” (Lines 58-65). Even a more extended description of the study background was added as in the following point. A related reference wad added as follows:

  • 12 - Sabato, A.; Dabetwar, S.; Kulkarni, N.N.; Fortino, G. Noncontact Sensing Techniques for AI-Aided Structural Health Monitoring: A Systematic Review. IEEE Sensors Journal 2023, 23(5), pp. 4672-4684.

  1. No background in seismic monitoring and techniques used. No classification of earthquakes. No pointing out problems like sudden and unexpected high amplitude earthquakes or so-called high activity earthquake swarm events that are potentially damaging to civil engineering structures.

Addressing comment # 6: The authors added to the introduction a more extensive background on seismic monitoring and techniques used as follows: “Several techniques of seismic monitoring are available that use several methods and sensors. They are generally intended to sense the vibration that occurs in a structure or possible changes in physical parameters related to structural health in consequence of a seismic event. In fact, several types of earthquakes may occur depending on the site where the structure is located. In particular, sudden and unexpected high amplitude earthquakes or so-called high activity earthquake swarm events can occur, potentially causing relevant damage to civil engineering structures [1].” (Lines 35-41). We also added a related reference as follows:

  • 1 - Scislo, L. High Activity Earthquake Swarm Event Monitoring and Impact Analysis on Underground High Energy Physics Research Facilities. Energies 2022, 15, 3705

  1. No alternative methods are mentioned that are also video-based but deal with the problem of the need for numerous measurement points. Some possibilities for other optical systems that can perform such measurements of even very big objects (buildings) include video recorders and virtual visual sensors (e.g. https://doi.org/10.1177/14759217145228), a digital image correlation (DIC) (e.g https://doi.org/10.1016/j.engstruct.2020.110551) and especially 3D Lase Doppler Vibrometry (e.g DOI: 10.3390/s23031263). In this last case, this is an optical system dedicated to vibration and seismic measurements but also possible to measure displacement and is used for quality control also in the case of civil engineering structures or building structural elements. Additionally, the system apart from displacement allows the measurement vibration (natural frequencies, mode shapes and damping) without the need to use accelerometers and another acquisition system. Please make the correction and the beginning of the introduction. Without it, there is no proper background to the research you are presenting.

Addressing comment # 7: The authors thank the reviewer for the suggestion about mentioning more alternative methods for completeness of state-of-the-art. The suggested methods were added in the introduction and related references were added in the reference section as follows: “Some video-based methods deal with the problem of the need for numerous measurement points. Among such methods, interesting possibilities are provided by optical systems that can perform such measurements of even very big objects (e.g. buildings), including video recorders, virtual vision sensors, digital image correlation (DIC), and 3D laser Doppler vibrometry [15,16]. In particular, an innovative system utilizing a 3D laser Doppler vibration measurement system capable was recently proposed. This system is able to extract the modal parameters of civil structures or building structural elements with interesting accuracy and reduced required time. However, 3D laser Doppler vibrometry still requires the use of quite expensive equipment with respect to low-cost cameras.” (Lines 77-85) The following references suggested by the reviewer were added:

- 15 - Ngeljaratan, L.; Moustafa, M.A. Structural health monitoring and seismic response assessment of bridge structures using target-tracking digital image correlation. Eng. Struct. 2020, 213, 110551.

- 16 - Scislo, L. Single-Point and Surface Quality Assessment Algorithm in Continuous Production with the Use of 3D Laser Doppler Scanning Vibrometry System. Sensors 2023, 23, 1263.

  1. Chapter 2 I good- I would advise however to enlarge Fig.1 – the texts are barely visible.

Addressing comment # 8: The authors thank the reviewer for the suggestion. The dimension of Fig.1 was enlarged according to reviewer’s suggestion.

  1. Chapter 3: Fig 2 and 3 please unify the drawings in case of style, colour etc. Especially Fig2b which Is of a low quality. Apart from that the chapter is very well written and clear with all the information needed.

Addressing comment # 9: The authors thank the reviewer for the suggestion. Fig 2 and 3 were replaced with new figures with unified style and colour in the drawings.

  1. Chapters 4 and 5 are good- no comments on the further improvements.

Addressing comment # 10: The authors appreciate the reviewer’s comment.

  1. The result presentation is very good. Would only ask, if possible, for a better quality of figures 21 and 22. Maybe change the grey marks, texts etc. to black so everything is clearly visible.

Addressing comment # 11: The authors thank the reviewer for the suggestion. In the figures 21 and 22 the grey marks and texts were turned to black for better quality and readability, according to reviewer’s suggestion.

  1. The conclusion is rather a discussion of the results. Please present stronger the novelty of the paper, application, future steps and improvements.

Addressing comment # 12: The authors thank the reviewer for the suggestion. We added some lines to the conclusions to stress the novelty of the paper, possible application, future steps and improvements as follows: “Therefore, the main novelty of this study is in pointing out a simple an low-cost procedure based on MM method to estimate with interesting accuracy the intensity and location of damage in a multi-story RC framed prototype that is a very representative example of existing buildings in Italy. Even if methods based on conventional contact sensors are still much more accurate and reliable, and provide the possibility to analyze the structural behaviour during the strong motion phase (nonlinear conditions), the presented vid-eo-based method showed the potential of providing a preliminary valuable estimate of the state and location of damage, almost in stationary conditions, that can help management and planning of further deepened monitoring campaign. Of course, the proposed procedure is applicable also to other construction typologies, such as masonry and steel constructions. The main limits are related to the frequency range of interest and amplitude of ambient vibration of the structure, so that, used equipment should be of appropriate cam-era speed and resolution. Future steps and improvements are essentially related to the main challenges of this kind of techniques: (i) the present study was conducted through a laboratory experiment with an almost real-scale physical model, but on-the-field applications still present more complicated site conditions, such as the presence of non-structural components (ii) the quality of the on-the-field captured images are significantly affected by the surrounding environment conditions, such as lighting stability, edge shelter by rain, snow and fog, ground vibration, etc. and (iii) the further developments of more effective algorithms to improve the reliability of results affected by the above mentioned disturbances are under way and may take advantage from recent advances in artificial intelligence and other innovative processing techniques.” (Lines 708-730)

  1. In conclusion: The paper is clear. The biggest advantage is the presence of both tests in laboratory conditions and simulations. The paper needs however a better presentation of novelty and especially adding some alternative techniques in the introduction and the background of the study.

Addressing comment # 13: The authors thank the reviewer for overall positive comment. For better presentation of novelty and especially adding some alternative techniques in the introduction and the background of the study, please, make reference to replies to comments 5-6 and 12.

  1. Due to some small flaws, the reviewer is marking the paper for minor revisions. Hope the authors will use some suggestions to improve this otherwise very interesting and good, in case of results presentation, paper.

Addressing comment # 14: The authors thank the reviewer for his/her positive comments and constructive suggestions. The authors made use of all reviewer’s comments to improve the paper quality.
